

# Contrasting microbial assembly patterns in the woody endosphere of hybrid and non-hybrid *Populus* trees

Kyle R. Grant[1,2], Steven W. Kembel[3], Sachin Naik[1] and Selvadurai Dayanandan[1]

[1] Department of Biology, Concordia University, Montréal, Québec, Canada
[2] Centre for Boreal Research, Northern Alberta Institute of Technology, Peace River, Alberta, Canada
[3] Département des Sciences biologiques, Université du Québec à Montréal, Montréal, Québec, Canada

## ABSTRACT

Endophytes asymptomatically infect virtually all plant species, yet little is known about endophyte community assembly and diversity within the woody tissues of forest trees. We utilised phylogenetic null models of alpha (ses.MNTD$_{ab}$ and ses.MPD$_{ab}$) and beta diversity (ses.$\beta$MNTD$_{ab}$ and ses.$\beta$MPD$_{ab}$) to infer the role of deterministic and stochastic ecological processes in structuring bacterial and fungal endophyte communities in the woody tissues of *Populus deltoides* and the naturally occurring *P. × jackii* hybrid complex (*P. deltoides × P. balsamifera*). Microbial communities were characterised through Illumina amplicon sequencing (MiSeq) of the ITS and 16S rRNA gene. We detected 227 fungal ASVs, which were mainly classified as Ascomycota (92.4%). Among the 667 bacterial ASVs detected, the majority were classified as phylum Actinobacteriota (47.6%) and Proteobacteria (44.9%). We predicted that hybridisation could lead to a host environment that applies weaker selective effects on microbial taxa due to variability in host chemical and morphological phenotypes. Although bacterial communities did not support our prediction, fungal assemblages of the hybrid host (*P. × jackii*) were more phylogenetically random within (ses.MNTD$_{ab}$) and between assemblages (ses.$\beta$MNTD$_{ab}$ and ses.$\beta$MPD$_{ab}$) then the non-hybrid (*P. deltoides*)— consistent with an increased role of stochastic community assembly processes and less selective host environment. Host identity had a large influence on fungal community composition (weighted UniFrac R$^2$ = 34%), which may result from the differences in fungal selection we detected between hosts. Conversely, host identity was a weaker predictor of bacterial composition (weighted UniFrac R$^2$ = 13%), which may reflect the more dominant role of stochasticity we detected in bacterial assembly. Our findings provide evidence that host hybridisation may alter fungal assembly processes and diversity within the woody endosphere, leading to more phylogenetically diverse associations both within and between the fungal assemblages of hybrid trees. More broadly, our results highlight how genetically diverse host populations may promote microbial biodiversity within forests and hybrid transition zones.

Corresponding author
Kyle R. Grant, kgrant@nait.ca

## INTRODUCTION

Microbes that asymptomatically establish within the internal tissues of plants (endosphere)—referred to as endophytes (*Wennström, 1994*; *Wilson, 1995*)—have gained considerable attention for their ability to influence the function and fitness of their plant hosts. Endophytes may influence host growth (*Ji, Gururani & Chun, 2014*; *Mayer, Dörr de Quadros & Fulthorpe, 2019*; *Varga et al., 2020*), susceptibility to pathogens (*Ji, Gururani & Chun, 2014*; *Mishra et al., 2018*; *Constantin et al., 2020*), nutrient acquisition, and tolerance to abiotic stress (*Baltruschat et al., 2008*; *Christian, Herre & Clay, 2019*; *Tiryaki, Aydın & Atıcı, 2019*; *Varga et al., 2020*; *Zhou et al., 2021*) and have been linked to ecosystem processes, such as the early stages of decomposition following leaf senescence (*Yuan & Chen, 2014*; *Szink et al., 2016*; *Guerreiro et al., 2018*) and wood decomposition (*Song et al., 2017*; *Cline et al., 2018*).

Endophytes form diverse communities within plants that vary across space and time (*Lau, Arnold & Johnson, 2013*; *Borruso et al., 2018*; *Gomes et al., 2018*; *Barge et al., 2019*; *Materatski et al., 2019*). Within forest trees, these communities are structured by a combination of factors, including the host (species or genotype) and microbial niche (host organ or tissue type) they inhabit, reflecting differences in the chemistry and morphology of these microbial environments (*Lau, Arnold & Johnson, 2013*; *Lamit et al., 2014*; *Cregger et al., 2018*; *Pellitier, Zak & Salley, 2019*; *Tellez et al., 2022*). Abiotic factors, including climate seasonality, temperature (*Gomes et al., 2018*; *Oita et al., 2021*), and precipitation (*Lau, Arnold & Johnson, 2013*; *Gomes et al., 2018*; *Oita et al., 2021*), also play a crucial role in structuring the diversity and composition of endophyte communities.

The aboveground surfaces and inner compartments of forest trees, collectively referred to as the phyllosphere, is a heterogeneous microbial environment whose surface area spans more than $10^8$ km$^2$ globally (*Morris & Kinkel, 2002*). This heterogeneity, combined with the ease of sampling aboveground plant tissues, makes the phyllosphere a tractable system for testing fundamental questions in ecology (see *Meyer & Leveau, 2012*) and elucidating the assembly rules structuring plant-associated microbial communities. Community assembly rules are most broadly classified as deterministic or stochastic: Deterministic processes are associated with ecological selection (*sensu Vellend, 2010*) that results from fitness differences between taxa within a given environment, including processes of competition, facilitation, and environmental filtering. Conversely, stochastic processes do not relate to fitness and include neutral processes of dispersal and ecological drift (*Stegen et al., 2013*; *Dini-Andreote et al., 2015*). In the phyllosphere, host structural and chemical factors that vary across plant species and compartments (*Lau, Arnold & Johnson, 2013*; *Borruso et al., 2018*; *Pellitier, Zak & Salley, 2019*; *Tellez et al., 2022*) may apply unique selective pressures on microbial taxa, contributing to deterministic community assembly. The influence of stochastic processes in the phyllosphere is thought to be most prevalent during the early stages of plant growth, as seen in young *Populus* trees (*Dove et al., 2021*) and the phyllosphere bacterial communities of *Arabidopsis thaliana* (*Maignien et al., 2014*) and perennial grass species (*Grady et al., 2019*). Stochastic processes may give way to deterministic as the initial microbial community is established, as better-adapted and more competitive microbes

replace early colonisers across time (*Emerson & Gillespie, 2008*; *Dini-Andreote et al., 2015*; *Dini-Andreote & Raaijmakers, 2018*).

Using data generated through amplicon sequencing (Illumina MiSeq), we examined bacterial- and fungal-endophytic community patterns in the aboveground woody tissues (twigs) of *P. deltoides* and the naturally occurring *P. × jackii* hybrid complex (*P. balsamifera × P. deltoides*), through a combination of phylogenetic and taxonomic alpha and beta diversity metrics. We aimed to examine how hybridisation may influence endophyte communities and assembly processes within the woody endosphere of this widely cultivated and ecologically important genus (*Cooke & Rood, 2007*), which has become a model organism for testing hypotheses regarding plant-microbe interaction in woody plants (see *Cregger et al., 2021*). Endophyte assembly in the phyllosphere of *Populus* is thought to be largely deterministic (*Bálint et al., 2013*; *Cregger et al., 2018*), with more stochastic processes dominating during the initial stages of plant growth (*Dove et al., 2021*). Although factors structuring the microbial diversity of *Populus* have been examined, it remains unclear how different ecological processes contribute to microbial assembly (but see *Dove et al., 2021*), especially within aboveground woody tissues. Understanding how host hybridisation may alter microbial assembly processes and diversity may be of interest, given the susceptibility of natural and cultivated hybrid *Populus* trees to wood-inhabiting pathogens of the genus *Septoria*, which currently limit the geographical extent of their cultivation (*Ostry, 1987*; *Newcombe & Ostry, 2001*; *Cregger et al., 2018*).

*Populus* hybrid zones can exceed the genetic diversity of parental species combined due to the ability of hybrid *Populus* to form backcrosses (*Whitham et al., 1999*). This diversity can lead to greater phenotypic variability within hybrid *Populus* populations (*Whitham et al., 1999*), as *Populus* genotypes are known to exhibit phenotypic plasticity and vary in their defensive compounds and physical traits (*Lindroth & St. Clair, 2013*; *Bandau et al., 2015*; *Liu & El-Kassaby, 2019*; *Bandau et al., 2021*). Breakdowns in chemistry, including the formation of defensive compounds, or changes in the physical structures of hybrid trees could lead to a less selective host environment, reducing the role of deterministic processes in structuring endophyte communities. We therefore hypothesised that hybridisation could shift the balance between stochastic and deterministic ecological processes within the woody endosphere of *Populus*.

Phylogenetic community patterns can provide insights into the role of stochastic *versus* deterministic processes in microbial assembly, under an assumption that the ecological preferences of microbial taxa are phylogenetically conserved (*Stegen et al., 2013*; *Dini-Andreote et al., 2015*)—that is, microbes that are close relatives (closer on a phylogeny) are more similar in their ecological preferences than distant relatives (*Webb et al., 2002*; *Cavender-Bares et al., 2009*; *Cadotte & Davies, 2016*). Under this assumption, when the host environment selects for microbial taxa with specific ecological preferences, the resulting microbial assemblage will be composed of closer or more distant relatives than expected (*i.e.,* compared to a null expectation of random assembly; see *Webb et al., 2002*; *Webb, 2000*). The observed phylogenetic structure of an assemblage will thus be less random as the balance between stochastic and deterministic processes shifts towards determinism. This approach of measuring phylogenetic structure relative to an expectation
of random assembly can be extended to the level of the host population (see *Stegen et al., 2013*; *Dini-Andreote et al., 2015*). For example, when individual host environments are homogeneous and apply strong selective effects on microbial taxa, each host environment may select for microbes with similar ecological preferences. If these ecological preferences are phylogenetically conserved, then the resulting microbial assemblages would be more similar (*i.e.*, phylogenetic beta diversity) than expected under random assembly (termed *homogenising selection*). Alternatively, if host environments are heterogeneous and apply strong selective effects, then each host environment selects for microbes with different ecological preferences, leading assemblages to be more phylogenetically dissimilar than expected (*variable selection*). Both patterns reflect an increasing role of determinism in microbial assembly (see *Dini-Andreote et al., 2015*) and more selective host environment.

We therefore predicted that the microbial assemblages associated with hybrid trees (*P. × jackii*) would be more phylogenetically random both within (alpha diversity) and between (beta diversity) assemblages compared to non-hybrids (*P. deltoides*), consistent with the hybrid host environment imposing weaker selection on microbial taxa and a greater role of stochastic ecological processes in microbial assembly. Woody endosphere bacterial and fungal communities are likely to respond differently to host environments, given differences in their life history strategies, dispersal abilities, and evolutionary histories (*Dove et al., 2021*). We thus contrasted community patterns across these two major groups to infer the role of host identity and community assembly processes in structuring their diversity. Our work may help to direct future studies examining the links between host hybridisation and the plant microbiome.

## MATERIALS & METHODS

### Sampling design

Twig samples were collected from natural tree stands located across two study sites in southern Quebec: Mont-Saint-Hilaire and Oka National Park (collection approved by the Société des établissements de plein air du Québec; authorisation number: PNO-2020-008), located ~70 km from each other. At each site, twigs were collected from two host taxa: *Populus deltoides* and the naturally occurring hybrid, *P. × jackii* (*P. balsamifera × P. deltoides*). Five trees of each host were sampled between July 20th and July 28th, 2020, at each of the two sites, resulting in a total of 10 trees per host. Five shaded branches measuring ~24 inches were excised from the phyllosphere of each tree at a height of ~10–20 ft from the forest floor. Cuttings were placed in plastic bags and transported to the laboratory on ice, where they were stored at 4 °C until processing.

### Sample processing

We processed all stored samples within 48 h of collection. Processing involved several steps: Asymptomatic twigs were first cut from tree branches using sterilised loppers and then rinsed with sterile double-distilled water and several drops of Tween 20 before surface sterilization. The twigs were then submerged and stirred vigorously in 70% ethanol for one minute, followed by 4% sodium hypochlorite for 10 min. We rinsed the samples between wash steps by submerging them in sterilised double-distilled water, with an additional five

rinses at the end of sterilisation to remove any residual chemicals. To assess the effectiveness of surface sterilisation, twigs were pressed into culture plates containing potato-dextrose agar and incubated at room temperature for 48 h. The sterilised twig samples were then stored in sterile falcon tubes at −80 °C until DNA extraction. We repeated these steps for each of the 20 twig samples.

## DNA extraction, library preparation, & sequencing

To homogenise each of the samples, a sterile scalpel was used to cut approximately 12 randomly sampled sterilised twigs into small pieces between 1.0 to 2.0 mm in size. The cut pieces were mixed, and 150 mg of tissue was collected in tubes for bead beating. Samples were bead-beaten for 90 s prior to DNA extraction. We extracted genomic DNA from samples using a DNeasy Plant Mini Kit (Qiagen) following the manufactures instructions.

To reduce contamination from host DNA, we chose to amplify the V5–V6 region of the 16S rDNA gene using the chloroplast discriminating primers, 799F/1115R. For fungal DNA amplification, we targeted the ITS1 rDNA gene using the universal fungal primers, ITS1F/ITS2. A two-step PCR approach was used to amplify both gene regions. PCR amplification of the V5–V6 region was performed in a 25 μL mixture containing three μL of DNA extract, 0.2 μM of both forward and reverse primer, 0.5 U of Phusion Hot Start II High-Fidelity DNA Polymerase (ThermoFisher), 1x of Phusion HF Buffer, 0.2 μM of dNTPs, and 3% DMSO. We used the same mixture for ITS1 amplification with the exception that four μL of DNA extract was included in the mixture.

PCR cycles for amplification of the V5-V6 region were performed for 30 s at 98 °C for initial denaturation, followed by 15 s at 98 °C, 36 cycles of 30s at 64 °C, 30 s at 72 °C, and 10 min at 72 °C for final extension. The same intermediate steps were used for ITS1 amplification, except 37 cycles were performed at 55 °C. A 2% agarose gel was used to visualise PCR products, which were normalised using a Just-a-Plate Normalisation Kit, following the manufacturer's protocols (CharmBiotech). Samples were pooled into V5-V6 and ITS1, and V5-V6 samples were purified with AMPure XP beads at a ratio of 0.75:1 (bead-to-DNA ratio; Beckman Coulter). ITS1 pools were purified three times with a 0.8:1 bead to DNA ratio to eliminate primer dimers. A PhiX control library (Illumina) was spiked into the amplicon pool, and Illumina MiSeq sequencing was performed using a MiSeq reagent kit V3 (paired-end 300 bp; Illumina).

## Bioinformatics & data processing

We utilised the Quantitative Insights Into Microbial Ecology pipeline (QIIME2 v.2020.11.1) for all data processing steps (*Bolyen et al., 2019*). Cutadapt was first used to trim primer regions from demultiplexed reads prior to denoising steps (*Martin, 2011*). Fungal and bacterial-derived datasets were denoised independently through the DADA2 algorithm to generate amplicon sequence variants (ASVs), remove chimeric and low-quality sequences, and merge paired-end sequences (*Callahan et al., 2016*). We used the default settings of DADA2 with the following exceptions: (1) low-quality tails were first removed from sequences (*Q* score <30) by trimming the ITS1 forward reads at 276 bp and the reverse reads at 239 bp. The 16S rRNA reads were trimmed at 281 and 217 bp, respectively; (2) We

used the pseudo-pooling parameter to increase the detection sensitivity of low-frequency ASVs.

Taxonomic assignment was then performed using the naive Bayes classifier available in the sklearn Python package using the SILVA database for 16S rRNA (*Quast et al., 2012*) and the UNITE database for ITS1 sequences (*Abarenkov et al., 2010*). Classified sequences were then filtered to remove contamination from non-target DNA, including chloroplast and plant DNA. Unassigned ASVs were also removed for downstream analysis. We detected three reads belonging to the genus *Modestobacter* and five reads belonging to an unidentified fungal ASV within bacterial and fungal negative control samples, respectively. The low number of reads detected indicates that our samples were free from contamination. We, therefore, removed the negative controls from subsequent analyses. Finally, we removed potentially spurious low-abundance ASVs that contained fewer than 10 sequences. Although there is no consensus on the most appropriate filtering threshold for phyllosphere amplicon sequencing studies, we chose a conservative filtering approach, as our analysis relied heavily on diversity comparisons, which may be sensitive to low-abundance taxa (*Nikodemova et al., 2023*). For example, *Nikodemova et al. (2023)* found that most sporadically detected operational taxonomic units generated through 16S rRNA sequencing occur at less than 10 copies, and removing such microbiota may improve the reliability of our microbial diversity estimates.

## Phylogenetic inference

Phylogenetic trees were constructed for bacterial and fungal communities using the FastTree 2 algorithm in QIIME2 with a midpoint root (*Price, Dehal & Arkin, 2010*); Figs. S1 & S2). However, the high variability of ITS sequences can lead to poor sequence alignments among disparate fungal lineages, resulting in unresolved phylogenies that may bias metrics of community phylogenetic structure (*Molina-Venegas & Roquet, 2014*). We, therefore, ran the fungal community phylogenetic analysis using two trees: the original tree generated with FastTree 2 and a second tree generated using a 'ghost tree-like' method (*Fouquier et al., 2016*), whereby extension trees inferred from ITS sequences were grafted onto a resolved backbone phylogeny (Fig. S3).

For the 'ghost tree-like' method, we used *Shen et al.*'s (*2020*) phylogeny of Ascomycota as a backbone tree, accessed through TreeHouse (*Steenwyk & Rokas, 2019*). To generate extension trees, we first grouped ASVs by Class and generated an ITS sequence alignment for each group using the MUSCLE algorithm (*Edgar, 2004*). We then generated an extension tree for each alignment, using the IQTREE algorithm (*Nguyen et al., 2014*) in QIIME2, and grafted the resulting trees onto the backbone tree at their respective crown node (*i.e.,* their Class node). An ITS extension tree was also generated for Basidiomycete sequences and grafted to the root of the backbone tree.

We focused our analysis on the fungal community results obtained with the FastTree 2 method, as we were only able to run the ghost tree analysis with ASVs identified to the class level, which excluded ∼36% of sequences; however, we report the results obtained with the ghost tree (Figs. S4–S7, Tables S1 & S2) and where they differ from those of the FastTree 2 method.

## Community analysis

Rarefaction curves were generated for bacterial and fungal samples independently (Fig. S8). To standardise sampling effort, we rarefied bacterial samples to 1,628 reads and fungal samples to 6,342 reads. These values represented the minimum resampling depths that would allow us to maintain all samples for subsequent analysis. To ensure adequate sequencing depth was achieved, we used Good's coverage index, which indicated that all samples had greater than 99% coverage (Fig. S9). Community analysis was performed with the rarefied dataset using the *phyloseq* (*McMurdie & Holmes, 2013*), *picante* (*Kembel et al., 2010*), and *vegan* (*Oksanen et al., 2020*) packages in R v.4.3.2 (*R Core Team, 2023*). We performed all community analyses independently on bacterial and fungal community data.

We first explored whether the taxonomic or phylogenetic diversity within samples (alpha diversity) differed between host taxa. Taxonomic measures included ASV richness, Simpson's dominance, Chao1, and Pilous evenness. We utilised *picante* (*Kembel et al., 2010*) to assess the phylogenetic dispersion of microbial communities (*i.e.,* if assemblages consist of taxa that are closer or more distant relatives than expected) by calculating the standardised effect size of mean pairwise distance (ses.MPD) and mean nearest taxon distance (ses.MNTD). Both indices represent complementary measures of dispersion, with ses.MPD placing greater emphasis on deeper branches connecting taxa in the phylogeny (basal patterns) and ses.MNTD shallower branches (terminal patterns; *Mazel et al., 2016*). We utilised the abundance-weighted versions of ses.MPD (ses.MPD$_{ab}$) and ses.MNTD (ses.MNTD$_{ab}$), which emphasise microbial taxa that are common within samples and, therefore, account for bias that may result from rare microbial ASVs. Both dispersion measures are independent of the species richness of samples and are expressed as $z$-scores relative to a null distribution of random community assembly. We computed ses.MPD$_{ab}$ and ses.MNTD$_{ab}$ using the null model argument *taxa.labels* and 1,000 randomisations. To assess potential differences in alpha diversity between hosts, an analysis of variance (ANOVA) was performed.

We then calculated the between-sample (beta diversity) equivalent of ses.MPD$_{ab}$ and ses.MNTD$_{ab}$—that is, ses.$\beta$MPD$_{ab}$ and ses.$\beta$MNTD$_{ab}$—following methods adapted from *Dini-Andreote et al. (2015)*. ses.$\beta$MPD$_{ab}$ and ses.$\beta$MNTD$_{ab}$ measure the phylogenetic distance separating ASVs in two assemblages (*Kembel et al., 2010*) and are expressed as $z$-scores relative to a null distribution of phylogenetic distances generated from random communities. The null distribution thus reflects the expected $\beta$MPD$_{ab}$ or $\beta$MNTD$_{ab}$ between assemblages when stochastic ecological processes dominate community assembly (*Dini-Andreote et al., 2015*). To calculate ses.$\beta$MPD$_{ab}$ and ses.$\beta$MNTD$_{ab}$, we first Hellinger transformed the community matrix (*Oksanen et al., 2020*) and calculated the observed $\beta$MPD$_{ab}$ and $\beta$MNTD$_{ab}$ between samples using the *comdistnt* and *comdist* functions, respectively. We then constructed null distributions using 1,000 randomly assembled communities generated by randomising the community matrix and ASV labels on the phylogeny using the *randomiseMatrix* function (with the *null.model* argument set to *richness*) and the *tipShuffle* function, respectively. Each random matrix was Hellinger transformed before calculating their respective $\beta$MPD$_{ab}$ or $\beta$MNTD$_{ab}$ pairwise distances.

ses.$\beta$MPD$_{ab}$ and ses.$\beta$MNTD$_{ab}$ were then calculated as follows:

$$ses.\beta = \beta_{obs} - \beta_{null}/sd(\beta_{null})$$

where *ses.*$\beta$ is ses.$\beta$MPD$_{ab}$ or ses.$\beta$MNTD$_{ab}$, $\beta_{obs}$ is the observed pairwise distance between samples ($\beta$MPD$_{ab}$ or $\beta$MNTD$_{ab}$), and $\beta_{null}$ and $sd(\beta_{null})$ are the mean and the standard deviation of the corresponding null distribution of each pairwise comparison, respectively. When ses.$\beta$MPD$_{ab}$ or ses.$\beta$MNTD$_{ab}$ diverge from the mean of the null distribution by more than two standard deviations ($z$-score $<-2$ or $>+2$), they significantly differ from $\beta$MPD$_{ab}$ or $\beta$MNTD$_{ab}$ expected under stochastic community assembly, reflecting the role of deterministic ecological processes. Conversely, ses.$\beta$MPD$_{ab}$ or ses.$\beta$MNTD$_{ab}$ $z$-scores within two standard deviations of the mean are consistent with stochastic processes dominating community assembly (*Stegen et al., 2013*; *Dini-Andreote et al., 2015*). Positive $z$-scores indicate that assemblages are more phylogenetically dissimilar than expected, reflecting variable selection ($z$-score $>+2$), while negative $z$-scores indicate that assemblages are more similar than expected, reflecting homogenising selection ($z$-score $<-2$; see *Dini-Andreote et al., 2015*).

To assess differences in community composition between hosts, we utilised the abundance-weighted taxonomic measure Bray–Curtis dissimilarity (*Bray & Curtis, 1957*) and its phylogenetic analog *weighted UniFrac* (*Lozupone & Knight, 2005*; *Lozupone, Hamady & Knight, 2006*). Distance matrices were generated using Hellinger-transformed community data and visualised through principal coordinates analysis (PCoA) to examine differences in the composition of host-associated microbial communities. We performed a permutational analysis of variance (PERMANOVA) on distance matrices to test for differences in community composition between host taxa and sites using the *adonis2* function in *vegan*. We also utilised a permutational analysis of multivariate dispersions (PERMDISP), available through the *betadisper* function, to test for differences in variance between the microbial communities of host taxa and sites using the same distance matrices (*Oksanen et al., 2020*).

To test for differences in the abundance of microbial taxa between host taxa (*i.e.,* differential abundance analysis), we used the Analysis of Compositions of Microbiomes with Bias Correction (ANCOM-BC) R package (*Lin & Peddada, 2020*). Given that ANCOM-BC accounts for differences in sequencing depth, we ran the analysis separately on filtered non-rarefied bacterial and fungal data. We used default arguments with the exception that we set the alpha threshold to 0.001 and *struc_zero* argument to TRUE. This argument performs a presence-absence test for microbial taxa that are only present within a single host. By default, the *ancombc* function excludes taxa from the analysis that are detected in fewer than 10% of samples, which for our analysis excluded taxa that were detected in only one sample.

## RESULTS

### Sequencing & filtering

Illumina sequencing generated $\sim$1.56 million raw reads (680,153 16S rRNA and 874,920 ITS1 reads). Denoising and quality filtering with DADA2 resulted in the retention of

356,641 and 360,050 reads from 16S rRNA and ITS1 libraries, respectively. 16S rRNA reads contained a high proportion of contamination (50.2% of reads), resulting from the presence of mitochondria and chloroplast DNA, which accounted for 28.5% and 21.7% of reads, respectively. ITS1 reads contained low amounts of *Populus* DNA contamination (~0.6% of reads) and lepidopteran DNA belonging to a single ASV (0.02%). Filtering out contaminants, unclassified reads, and low-abundance ASVs (<10 sequences; *Nikodemova et al., 2023*) resulted in 355,245 fungal and 176,528 bacterial reads, belonging to 227 and 667 ASVs, respectively. The number of sequences per sample ranged from 1,628 to 35,003 for bacterial samples and 6,342 to 39,711 for fungal samples. We rarefied samples to a sequencing depth of 1,628 and 6,342 for bacterial and fungal samples, respectively (Fig. S8). These rarefied data were used for all subsequent steps in our analysis. No differences in the qualitative patterns or significance of our results were detected between resampled runs of the analysis. We present the results of a single run with rarefied data.

## Alpha & beta diversity analysis

The bacterial endophytic communities of twigs were, on average, more diverse and exhibited more even species distributions than fungal communities, although no differences in taxonomic diversity were detected between host taxa (Fig. S10). Measures of basal and terminal phylogenetic dispersion indicated that both bacterial and fungal communities trended towards phylogenetic clustering (*i.e.,* negative $z$-scores), although some individual host plants harboured microbial assemblages with weak to random phylogenetic structure (*i.e., z*-scores $\approx 0$). We detected no significant differences in phylogenetic dispersion between the microbial communities of host taxa, except for the fungal community of *P. × jackii*, which was less terminally clustered (ses.MNTD$_{ab}$) than *P. deltoides* (Fig. 1).

Phylogenetic beta diversity patterns revealed that microbial assembly was driven by a combination of ecological processes. Stochastic processes tended to be more prevalent in bacterial assembly; however, deterministic processes were more common within the bacterial community of *P. × jackii* than *P. deltoides* when measured through ses.$\beta$MNTD$_{ab}$ (53.3% *versus* 24.5% of pairwise comparisons, respectively; Fig. 2). Conversely, fungal assembly was characterised mainly by deterministic processes, although stochastic processes were more dominant in the fungal community of *P. × jackii* than *P. deltoides* when measured through ses.$\beta$MNTD$_{ab}$ (80.0% *versus* 20.0% of pairwise comparisons, respectively) and ses.$\beta$MPD$_{ab}$ (44.4% *versus* 4.4%, respectively; Fig. 2). The fungal community of *P. deltoides* displayed a strong signature of determinism in the form of homogenising selection (80.0% of ses.$\beta$MNTD$_{ab}$ and 95.6% of ses.$\beta$MPD$_{ab}$ pairwise comparisons; Fig. 2).

Visualisation of community composition through PCoA plots revealed that host-associated communities formed distinct clusters within multivariate ordination space through both weighted UniFrac (Fig. 3) and Bray–Curtis distance measures computed on Hellinger-transformed community data (Fig. S11). Results from the PERMANOVA (using the same distance matrices) indicated that variance in the fungal and bacterial community was explained by host identity, while sampling site and the interactions between sampling site and host identity were not significant predictors (Table 1). The fungal community of *P.*

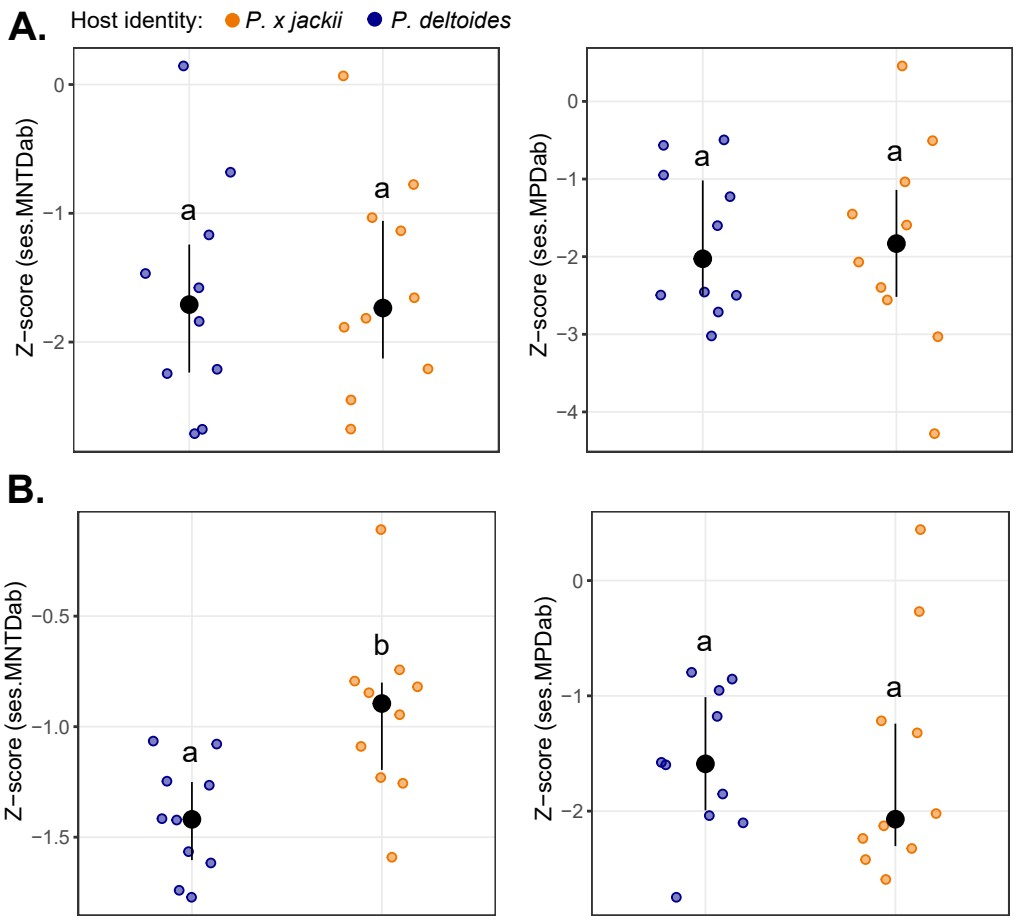

**Figure 1 Comparisons of phylogenetic dispersion (ses.MNTD$_{ab}$ and ses.MPD$_{ab}$) of microbial communities associated with the twig endobiome of *P. deltoides* and *P. × jackii*.** The phylogenetic dispersion of host-associated (A) bacterial and (B) fungal communities are displayed. Letters within plots indicate significant differences between host taxa (ANOVA at 95% confidence). The interquartile range and median are displayed (black).

× *jackii* had greater variance in taxonomic and phylogenetic community composition (*i.e.,* weighted UniFrac and Bray–Curtis distances) than *P. deltoides*, as indicated by PERMDISP (Fig. 3 & Table 2). However, we did not detect significant differences in variance through PERMDISP analysis with the ghost tree, suggesting that fungal community variance among *P. × jackii* is primarily driven by ASVs that are not identified to the class level (Table S2). No significant differences in bacterial community variance were detected between host taxa (Fig. 3 & Table 2).

## Bioindicator detection

Fungal endophytic samples were dominated by ASVs belonging to Ascomycota (92.4%), with Basidiomycota present at lower abundances (0.2%). The remaining ASVs were unassigned fungal sequences (7.4%). At the class level, Dothideomycetes were the most abundant ASVs (49.1%), followed by Eurotiomycetes (30.2%), Orbiliomycetes (1.0%),
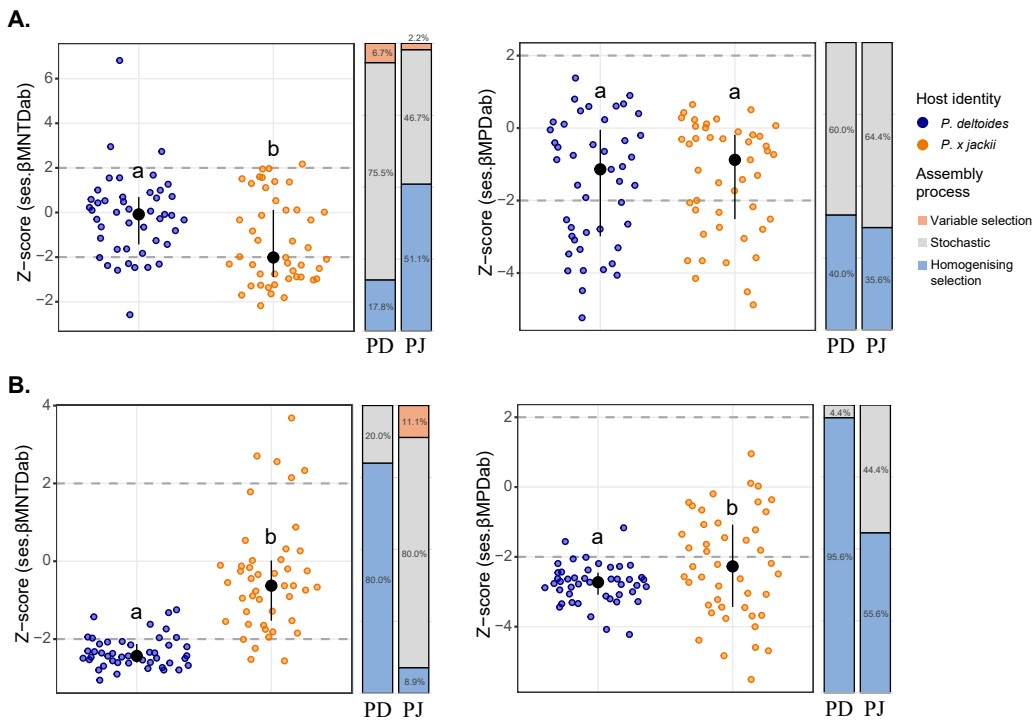

**Figure 2** Comparisons of phylogenetic beta diversity (ses.$\beta$MNTD$_{ab}$ and ses.$\beta$MPD$_{ab}$) and assembly processes across microbial communities associated with the twig endobiome of *P. deltoides* and *P. × jackii.* The phylogenetic beta diversity of host-associated (A) bacterial and (B) fungal communities are displayed. Points represent pairwise comparisons between samples, expressed as $z$-scores relative to a null model of random community assembly (see Methods). Letters within plots indicate significant differences between host taxa (ANOVA at 95% confidence), and the interquartile range and median are displayed (black). Grey dashed lines indicate two standard deviations from the mean of the null distribution, beyond which pairwise comparisons are characterised by variable ($z$-score $> +2$) or homogenising selection ($z$-score $< -2$). Bar charts indicate the percent of pairwise comparisons characterised by different community assembly processes for *P. deltoides* (PD) and *P. × jackii* (PJ).

Leotiomycetes (0.7%), Saccharomycetes (0.6%), and Agaricomycetes (0.2%), with ~36% of sequences not identified to the class level (Fig. 4). Through ANCOM-BC analysis, we detected a greater relative abundance of phylum Basidiomycota in *P. × jackii* samples (which were absent in *P. deltoides*), as well as the classes Agaricomycetes, Eurotiomycetes, and Orbiliomycetes, while the class Sordariomycetes was more abundant in *P. deltoides* samples (Fig. 4 & Table S3). We detected many differentially abundant fungal ASVs between host taxa; the most abundant of which belonged to genus *Neoconiothyrium* (Dothideomycetes), which was detected in greater abundances in *P. deltoides*, while ASVs belonging to genus *Xenocylindrosporium* (Eurotiomycetes), *Orbilia* (Orbiliomycetes), and *Knufia* (Eurotiomycetes) were more abundant in *P. × jackii.* Additional differentially abundant genera and ASVs were detected; however, their relative abundances were low (Table 3, Tables S3 & S4).

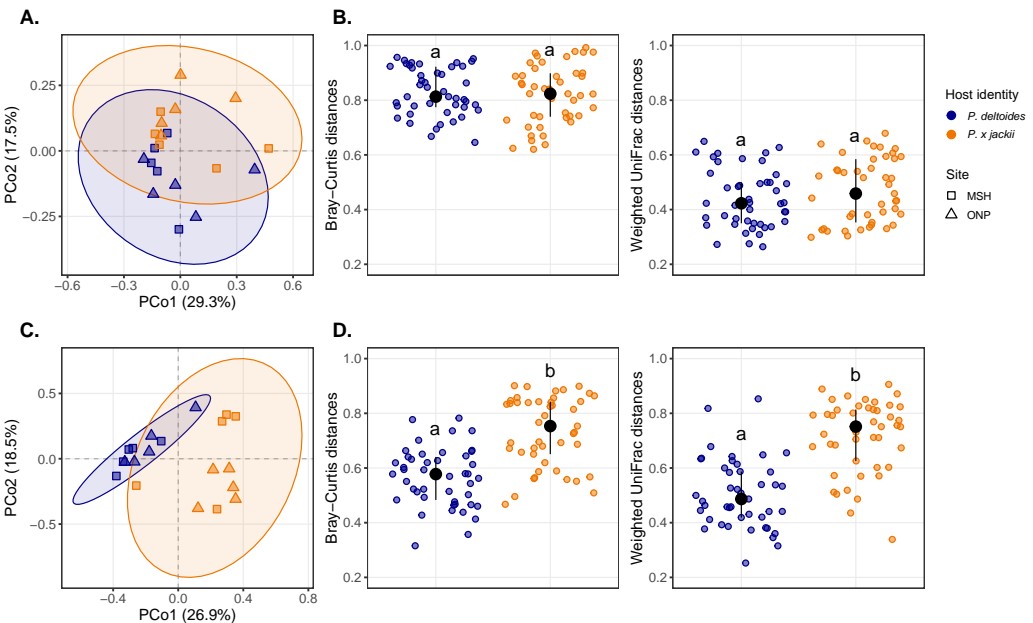

**Figure 3** **Compositional analysis of microbial communities associated with the twig endobiome of *P. deltoides* and *P. × jackii*.** Principal coordinates analysis (PCoA) of (A) bacterial and (C) fungal community composition based on weighted UniFrac distances computed with Hellinger transformed data are displayed. Interindividual taxonomic (Bray–Curtis distance) and phylogenetic beta diversity (weighted UniFrac distance) of (B) bacterial and (D) fungal communities computed with Hellinger transformed data are also presented. Letters within plots indicate significant differences between host taxa (ANOVA at 95% confidence). The interquartile range and median are displayed (black).

**Table 1** **PERMANOVA results for bacterial and fungal communities associated with the twig endobiome of *P. deltoides* and *P. × jackii*.**

| Model | Bray–Curtis distance | | | Weighted UniFrac distance | | |
|---|---|---|---|---|---|---|
| | *F*-value | $R^2$ | *p*-value | *F*-value | $R^2$ | *p*-value |
| **Bacterial community** | | | | | | |
| *Host identity* | **3.287** | **0.153** | **<0.001** | **2.575** | **0.128** | **0.012** |
| *Sites* | 1.058 | 0.049 | 0.319 | 0.450 | 0.022 | 0.932 |
| *Interaction* | 1.075 | 0.050 | 0.297 | 1.110 | 0.055 | 0.334 |
| **Fungal community** | | | | | | |
| *Host identity* | **12.417** | **0.392** | **<0.001** | **9.417** | **0.341** | **<0.001** |
| *Site* | 1.852 | 0.058 | 0.090 | 1.059 | 0.038 | 0.330 |
| *Interaction* | 1.422 | 0.045 | 0.178 | 1.134 | 0.041 | 0.307 |

**Notes.**
The community matrices were Hellinger transformed prior to computing Bray–Curtis and weighted UniFrac distances. Bolded values highlight significant factors ($p < 0.05$).

The most dominant bacteria of endophytic twig communities belonged to phylum Actinobacteriota (47.6%) and Proteobacteria (44.9%), with smaller abundances of Bacteroidota (5.3%). ASVs belonging to Abditibacteriota, Acidobacteriota, Armatimonadota, Bdellovibrionota, Deinococcota, Firmicutes, Fusobacteriota, Gemmatimonadota, and

**Table 2** PERMDISP results for bacterial and fungal communities associated with the twig endobiome of *P. deltoides* and *P.* × *jackii.*

| Model | Bray–Curtis distance | | | Weighted UniFrac distance | | |
|---|---|---|---|---|---|---|
| | Sum Sq | *F*-value | *p*-value | Sum Sq | *F*-value | *p*-value |
| **Bacterial community** | | | | | | |
| *Host identity* | <0.001 | 0.081 | 0.773 | 0.004 | 0.194 | 0.664 |
| *Sites* | <0.001 | 0.126 | 0.721 | 0.010 | 0.473 | 0.509 |
| *Interaction* | 0.013 | 0.612 | 0.615 | 0.038 | 0.629 | 0.627 |
| **Fungal community** | | | | | | |
| *Host identity* | **0.055** | **6.439** | **0.018** | **0.121** | **13.438** | **0.002** |
| *Site* | 0.001 | 0.168 | 0.686 | 0.004 | 0.244 | 0.625 |
| *Interaction* | 0.061 | 1.507 | 0.260 | 1.102 | 2.060 | 0.155 |

**Notes.**
The community matrices were Hellinger transformed prior to computing Bray–Curtis and weighted UniFrac distances. Bolded values highlight significant factors ($p < 0.05$).

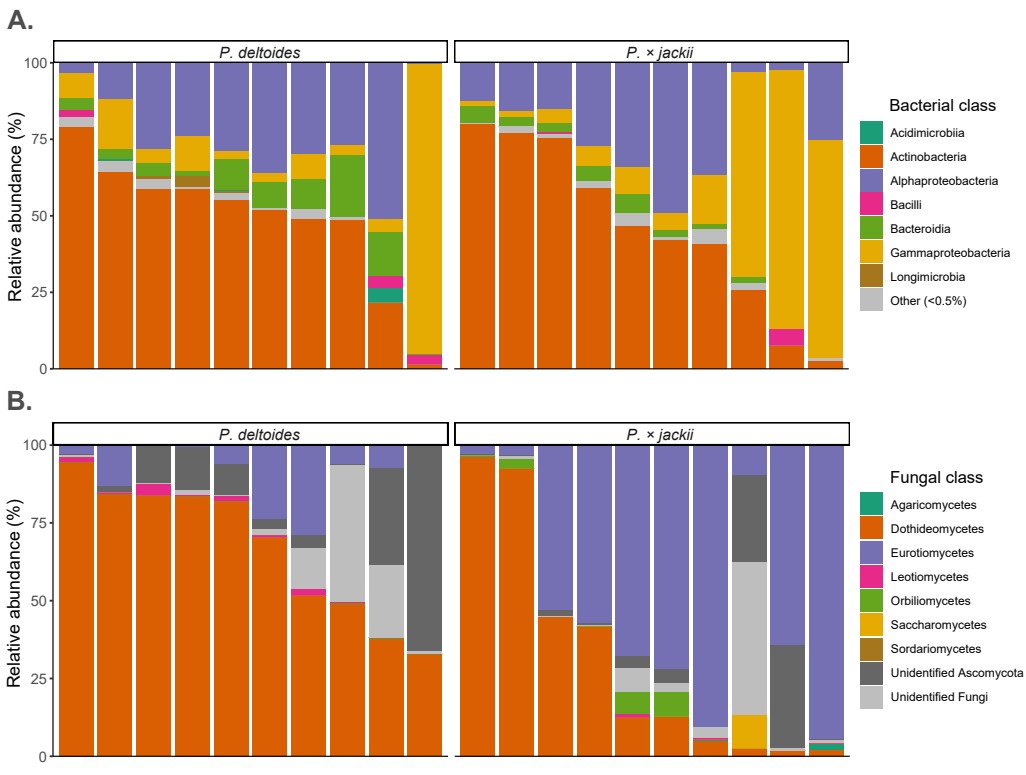

**Figure 4** **Relative abundance of microbial communities associated with the twig endobiome of *P. deltoides* and *P.* × *jackii.*** Data were aggregated to display (A) bacterial and (B) fungal classes among individual trees of *P. deltoides* and *P.* × *jackii*. Bacterial classes with relative abundances less than 0.5% are grouped as "Other".

Myxococcota were also detected at relative abundances of less than 1%. Through ANCOM-BC analysis, we detected a greater relative abundance of phylum Gemmatimonadota in *P. deltoides* samples, while Acidobacteriota were in greater abundances in *P.* × *jackii* samples.

Grant et al. (2025), *PeerJ*, DOI 10.7717/peerj.20073
**Table 3   Relative abundance of the twenty most abundant fungal ASVs associated with the twig endobiome of *P. deltoides* and *P. × jackii*.**

| *Populus deltoides* | | | *Populus × jackii* | | |
|---|---|---|---|---|---|
| **Class** | **Genus** | **ASV (%)** | **Class** | **Genus** | **ASV (%)** |
| **Dothideomycetes** | *Neoconiothyrium* | **34.46** | **Eurotiomycetes** | *Xenocylindrosporium* | **34.84** |
| **Dothideomycetes** | *Endosporium* | **10.25** | Dothideomycetes | Unidentified dothideales | 14.79 |
| **Unidientified ascomycota** | **Unidentified** | **10.19** | **Dothideomycetes** | *Neoconiothyrium* | **8.45** |
| Unidentified fungi | Unidentified | 6.16 | **Eurotiomycetes** | *Xenocylindrosporium* | **6.31** |
| **Eurotiomycetes** | *Xenocylindrosporium* | **5.76** | **Eurotiomycetes** | *Xenocylindrosporium* | **5.47** |
| **Dothideomycetes** | *Neoconiothyrium* | **4.71** | **Unidentified fungi** | **Unidentified** | **4.82** |
| **Dothideomycetes** | *Neoconiothyrium* | **4.01** | **Unidientified ascomycota** | **Unidentifed** | **4.68** |
| **Dothideomycetes** | *Neoconiothyrium* | **3.23** | **Dothideomycetes** | *Endosporium* | **2.56** |
| **Dothideomycetes** | *Phoma* | **3.04** | **Dothideomycetes** | *Endosporium* | **1.96** |
| **Unidientified ascomycota** | **Unidentified** | **2.43** | **Unidientified ascomycota** | **Unidentified** | **1.33** |
| Dothideomycetes | Unidentified dothideales | 2.01 | **Orbiliomycetes** | *Orbilia* | **1.17** |
| Dothideomycetes | Unidentified dothideales | 1.10 | **Eurotiomycetes** | *Xenocylindrosporium* | **1.03** |
| **Eurotiomycetes** | **Unidentified phaeomoniellales** | **1.07** | **Eurotiomycetes** | **Unidentified Phaeomoniellales** | **0.89** |
| Leotiomycetes | *Phialocephala* | 1.05 | Saccharomycetes | *Nakazawaea* | 0.72 |
| Unidentified fungi | Unidentified | 0.80 | **Unidentified fungi** | **Unidentified** | **0.67** |
| **Unidientified ascomycota** | **Unidentified** | **0.79** | **Eurotiomycetes** | *Xenocylindrosporium* | **0.52** |
| Dothideomycetes | *Endosporium* | 0.76 | **Eurotiomycetes** | *Knufia* | **0.49** |
| Dothideomycetes | *Elsinoe* | 0.75 | **Dothideomycetes** | **Unidentified myriangiales** | **0.41** |
| Unidentified fungi | Unidentified | 0.69 | Saccharomycetes | *Candida* | 0.34 |
| **Dothideomycetes** | *Neoconiothyrium* | **0.56** | **Eurotiomycetes** | **Unidentified chaetothyriales** | **0.33** |

**Notes.**

Bolded text indicates ASVs that were differentially abundant between host taxa and detected in more than on sample (ANCOM-BC analysis; adjusted $p < 0.001$).

The classes Acidimicrobiia, Longimicrobia, and Polyangia were more abundant within *P. deltoides*, while Acidobacteriae and Armatimonadia were more abundant in *P. × jackii* (Fig. 4 & Table S5). We detected many additional differentially abundant ASVs and taxa at lower taxonomic levels (Table 4, Tables S5 & S6).

## DISCUSSION

We examined taxonomic and phylogenetic alpha and beta diversity patterns of bacterial and fungal endophytic communities within the aboveground woody tissues of two naturally occurring hosts, *Populus deltoides* and *P. × jackii* (*P. balsamifera × P. deltoides*). Host bacterial communities did not support our prediction that hybridisation would lead to a less selective host environment, as we detected no signature of increased stochasticity through phylogenetic alpha or beta diversity metrics. Host identity explained little of the variation in bacterial communities (weighted UniFrac $R^2 = 13\%$), which may reflect the generally dominant role of stochasticity in bacterial assembly that we detected. Fungal communities supported our prediction, as fungal assemblages associated with the hybrid host (*P. × jackii*) exhibited a more random terminal phylogenetic structure (ses.MNTD$_{ab}$), and stochastic processes had a greater influence on community assembly. The hybrid host environment may thus apply weaker selective pressures on fungal lineages, contributing to the greater divergence in fungal community composition we detected across host environments (weighted UniFrac $R^2 = 34\%$). Our findings provide evidence that hybridisation can lead to more phylogenetically diverse fungal associations within and between woody endosphere assemblages, resulting from a less selective host environment due to a weaker role of deterministic ecological processes in fungal assembly. This reduction in fungal selection may explain the susceptibility of hybrid *Populus* trees to wood-inhabiting pathogens (*Ostry, 1987*; *Newcombe & Ostry, 2001*) and has broader implications for the link between host genetic diversity and microbial biodiversity within forests and hybrid transition zones.

The differences in phylogenetic structure and ecological processes we detected between the fungal communities of host taxa were most pronounced when measured through terminal metrics of phylogenetic alpha and beta diversity (ses.MNTD$_{ab}$ and ses.$\beta$MNTD$_{ab}$), although we also detected differences through ses.$\beta$MPD$_{ab}$. This larger signal in terminal metrics would be consistent with processes acting on fungal traits or ecological preferences that are shallow on the phylogeny, reflecting more recent adaptation among lineages, assuming phylogenetic conservatism (*Mazel et al., 2016*; *Webb et al., 2002*). For endophytes, such traits could include those that alter their ability to bypass physical barriers on the surface of the host plant or differences in their metabolic requirements. However, non-random phylogenetic patterns may also result from competitive interactions when traits that make species stronger competitors within an environment are phylogenetically conserved (see *Davies, 2021*). Overall, our findings suggest that the hybrid host environment applies weaker selective effects on microbial taxa, resulting in a greater diversity of fungal lineages within individual trees and between individuals at the host population level. *Populus* genotypes are known to vary in their concentrations of antifungal compounds

*Peer*J

**Table 4** **Relative abundance of the twenty most abundant bacterial ASVs associated with the twig endobiome of *P. deltoides* and *P.* × *jackii*.**

| Populus deltoides | | | Populus × jackii | | |
|---|---|---|---|---|---|
| **Class** | **Genus** | **ASV (%)** | **Class** | **Genus** | **ASV (%)** |
| *Gammaproteobacteria* | Unidentified enterobacterales | 8.89 | ***Actinobacteria*** | ***Actinoplanes*** | **10.34** |
| *Actinobacteria* | Unidentified microbacteriaceae | 4.91 | *Actinobacteria* | Unidentified micromonosporaceae | 9.42 |
| *Actinobacteria* | *Modestobacter* | 4.13 | *Gammaproteobacteria* | *Xanthomonas* | 5.17 |
| *Actinobacteria* | *Modestobacter* | 2.95 | *Gammaproteobacteria* | Unidentified burkholderiaceae | 3.24 |
| ***Actinobacteria*** | ***Quadrisphaera*** | **2.76** | *Gammaproteobacteria* | *Halotalea* | 3.22 |
| *Alphaproteobacteria* | Unidentified xanthobacteraceae | 2.73 | ***Alphaproteobacteria*** | **1174-901-12** | **3.05** |
| ***Actinobacteria*** | ***Kineococcus*** | **2.52** | *Actinobacteria* | *Curtobacterium* | 2.97 |
| *Actinobacteria* | *Actinomycetospora* | 2.33 | *Actinobacteria* | *Actinomycetospora* | 2.27 |
| ***Actinobacteria*** | ***Modestobacter*** | **2.24** | *Actinobacteria* | *Actinoplanes* | 2.04 |
| ***Actinobacteria*** | ***Nocardioides*** | **2.05** | *Gammaproteobacteria* | *Pseudomonas* | 1.97 |
| *Actinobacteria* | *Klenkia* | 1.87 | *Alphaproteobacteria* | *Sphingomonas* | 1.97 |
| *Actinobacteria* | *Modestobacter* | 1.81 | *Actinobacteria* | *Kineosporia* | 1.70 |
| ***Alphaproteobacteria*** | ***Corticibacterium*** | **1.73** | ***Actinobacteria*** | **Unidentified kineosporiaceae** | **1.64** |
| *Actinobacteria* | *Curtobacterium* | 1.70 | ***Actinobacteria*** | ***Modestobacter*** | **1.63** |
| *Actinobacteria* | *Quadrisphaera* | 1.47 | ***Actinobacteria*** | ***Quadrisphaera*** | **1.55** |
| *Alphaproteobacteria* | *Bradyrhizobium* | 1.44 | *Gammaproteobacteria* | Unidentified oxalobacteraceae | 1.47 |
| *Alphaproteobacteria* | *Methylobacterium-Methylorubrum* | 1.16 | *Actinobacteria* | *Actinoplanes* | 1.20 |
| ***Actinobacteria*** | ***Frigoribacterium*** | **1.04** | *Alphaproteobacteria* | *Sphingomonas* | 1.15 |
| *Actinobacteria* | *Frondihabitans* | 0.89 | *Gammaproteobacteria* | *Pseudomonas* | 1.15 |
| *Bacteroidia* | *Hymenobacter* | 0.85 | ***Gammaproteobacteria*** | **Unidentified burkholderiaceae** | **1.08** |

Notes.
Bolded text indicates ASVs that were differentially abundant between host taxa and detected in more than one sample (ANCOM-BC analysis; adjusted $p < 0.001$).

and secondary metabolites (*Lindroth et al., 2002*; *Chen et al., 2009*), and differences in their concentration or composition between genetically diverse host individuals may contribute to heterogeneity in the host environment (*Whitham et al., 1999*). Such heterogeneity may contribute to the relatively weaker selection and greater compositional variance (PERMDISP) we observed among *P. × jackii* fungal assemblages. Previous work has also linked bark fungal community variance among tree species to chemical properties, such as pH, nitrogen, and total phenolic content (*Pellitier, Zak & Salley, 2019*).

Bacterial phylogenetic beta diversity was mainly attributable to stochastic ecological processes (although deterministic processes were more prevalent in *P. ×jackii* when measured through ses.$\beta$MNTD$_{ab}$). Our finding that determinism may have a weaker role in bacterial assembly is consistent with previous findings from the leaf and root bacteriome of young *Populus* trees, where determinism explained ∼8% of pairwise comparisons between bacterial assemblages, compared to ∼54% of fungal comparisons (*Dove et al., 2021*). Although bacterial assembly was more stochastic, we note that deterministic processes were still important, and bacterial assemblages were, on average, phylogenetically clustered when measured through alpha diversity metrics.

Host identity had a weak influence on bacterial community composition (Bray–Curtis $R^2 = 15\%$; weighted UniFrac $R^2 = 13\%$), suggesting that bacterial assemblages are primarily composed of microbes that are more generally adapted to endophytic lifestyles rather than specialised to specific host environments. *Frank, Saldierna Guzmán & Shay (2017)* suggested that bacterial endophytes are often horizontally transferred generalists, with regional species pools being more important in structuring endophyte communities than host genotypes (*Yeoh et al., 2017*). This suggestion is supported by our findings and those of *Cregger et al. (2018)*, who found a similar weak influence of host identity on the twig bacterial community of three-year-old *Populus* trees (Bray–Curtis $R^2 = 10\%$). This similarity in the role of host identity between mature and juvenile *Populus* trees suggests that the host-related deterministic processes that structure bacterial communities in the woody endosphere remain relatively stable across the host's lifespan. Initial bacterial assembly may, therefore, be important for establishing the bacterial endobiome, with the host environment imposing relatively weak selective effects on bacterial taxa.

Host identity had a larger influence on fungal community structure (Bray–Curtis $R^2 = 39\%$; weighted UniFrac $R^2 = 34\%$). Host genotypes are known to be a significant determinant of leaf fungal endophytic composition in *Populus* species, suggesting plant-fungal coevolutionary relationships (*Bálint et al., 2013*; *Cregger et al., 2018*; *Dove et al., 2021*); however, previous findings have demonstrated a weak influence of host identity on twig fungal communities (Bray–Curtis $R^2 = 9\%$; *Cregger et al., 2018*). This discrepancy in our results may arise from *Cregger et al. (2018)* utilising three-year-old trees for their analysis, as stochastic processes may have a greater influence on community assembly during the initial stages of tree growth (*Dove et al., 2021*). Given that the twig endosphere is a non-transient environment, community succession is likely to occur across the host's lifespan. Our finding that host identity is a strong predictor of fungal communities in mature *Populus* trees could suggest that host-adapted fungi accumulate in the woody endosphere as the host matures—perhaps replacing more generalist endophytes. Alternatively, the

differences in fungal assembly processes we detected between hosts could drive divergence in their communities, as non-hybrids select for host-adapted fungi while hybrids associate with a broader range of stochastically assembled generalist endophytes.

ANCOM-BC analysis supports the role of host selective mechanisms in structuring the microbial endophytic community of woody tissues, with several bacterial and fungal bioindicators detected across host taxa. Most of the dissimilarity in host-associated fungal communities resulted from differences in the relative abundance of dominant ASVs belonging to the genus *Neoconiothyrium* (class Dothideomycetes), which were more abundant in the woody tissues of *P. deltoides*, and genus *Xenocylindrosporium* (class Eurotiomycetes), which dominated *P. × jackii*—although several other less abundant genera were also differentially abundant. Our findings suggest that interspecific variance in the twig fungal endophytic community reflects differences in a relatively small number of genera. We detected no differences in bacterial or fungal community structure between sampling sites (PERMANOVA), suggesting that microbes are filtered from a similar species pool at the spatial scale of our study (∼70 km between sampling sites).

## CONCLUSIONS

Our results highlight how woody endosphere environments may impose different selective effects on bacterial and fungal microbiota, reflecting differences in the role of deterministic *versus* stochastic community assembly processes. We found that host hybridisation may lead to more phylogenetically diverse fungal associations both within and between the aboveground woody tissues of individual trees. This increase in the diversity of fungal associations among hybrid trees reflects a less selective host environment and a weaker influence of determinism in fungal assembly. Our work also demonstrates that endophytic fungal and bacterial communities in the woody endosphere of *Populus* trees are structured primarily by host identity rather than geography at the scale of our analysis (sample sites were ∼70 km apart). Differences in fungal community composition between hosts are likely driven by the differences in fungal assembly processes we detected between hybrid and non-hybrid trees—as the non-hybrid host selects for specific fungal lineages, while the hybrid associates with a phylogenetically broader range of stochastically assembled endophytes. The weak influence of host identity on bacterial community composition may reflect the dominance of stochastic processes in bacterial assembly that we detected, as both hosts harbour similar communities of bacteria that are randomly assembled from the regional species pool.

Although it is difficult to know the consequence of these fungal community differences for host plant health and fitness, more phylogenetically diverse fungal associations in hybrid trees may contribute to host adaptability under novel environmental conditions. Alternatively, by imposing more relaxed selection pressures on fungal lineages, hybrid *Populus* trees may be more susceptible to colonisation by a broader range of fungal taxa, such as pathogenic fungi. Understanding how community assembly processes in woody tissues diverge from non-hybrids can help guide genetic breeding programs and biocontrol interventions aimed at mitigating disease outbreaks, such as fungal canker

diseases, which commonly infect hybrid *Populus* trees (*i.e.*, *Septoria* species; *Ostry, 1987*; *Newcombe & Ostry, 2001*; *Cregger et al., 2018*). Furthermore, our work highlights the role that host genetic diversity can have in structuring the assembly and diversity of microbial communities within forests and hybrid transition zones.

We emphasise that primer selection can bias community composition in amplicon sequencing studies (*Johnson et al., 2019*) and that the endobiome is inherently dynamic and known to vary across temporal dimensions (*Borruso et al., 2018*; *Barge et al., 2019*; *Materatski et al., 2019*). As our work represents a single time point, it is important to consider that community patterns may vary across sampling dates. Furthermore, many of the fungal ASVs we detected were not identified to the class level (∼36%), which highlights the underrepresentation of woody endosphere microbiota in public sequencing databases. Our findings suggest that future work should investigate the influence of host genetic diversity on chemical and physical traits that may influence microbial community assembly. Further studies incorporating host traits and microbial assembly models could elucidate the role of different host factors in structuring microbial assembly and biodiversity within the phyllosphere.

## ACKNOWLEDGEMENTS

We thank David Maneli (McGill University - Gault Nature Reserve, Mont-Saint-Hilaire) and Société des établissements de plein air du Québec for support with fieldwork and access to sampling sites and Geneviève Bourret for their assistance with molecular laboratory work.

### Funding

Financial support for this project was provided by the Natural Sciences and Engineering Research Council of Canada (NSERC) Discovery Grant to Selvadurai Dayanandan. The funders had no role in study design, data collection and analysis, decision to publish, or preparation of the manuscript.

### Grant Disclosures

The following grant information was disclosed by the authors:
Natural Sciences and Engineering Research Council of Canada (NSERC) Discovery Grant.

### Competing Interests

The authors declare there are no competing interests.

### Author Contributions

- Kyle R. Grant conceived and designed the experiments, performed the experiments, analyzed the data, prepared figures and/or tables, authored or reviewed drafts of the article, and approved the final draft.
- Steven W. Kembel conceived and designed the experiments, authored or reviewed drafts of the article, and approved the final draft.

- Sachin Naik performed the experiments, authored or reviewed drafts of the article, and approved the final draft.
- Selvadurai Dayanandan conceived and designed the experiments, authored or reviewed drafts of the article, and approved the final draft.

### Field Study Permissions

The following information was supplied relating to field study approvals (i.e., approving body and any reference numbers):

Sample collection was approved by the Société des établissements de plein air du Québec (authorisation number: PNO-2020-008).

### Data Availability

DNA sequences are available at the NCBI Sequence Read Archive: PRJNA1159841.

Data and code are available at Figshare: Grant, Kyle (2025). Data and code from "Contrasting microbial assembly patterns in the woody endosphere of hybrid and non-hybrid Populus trees". figshare. Dataset. https://doi.org/10.6084/m9.figshare.26961913.v1

### Supplemental Information

Supplemental information for this article can be found online at http://dx.doi.org/10.7717/peerj.20073#supplemental-information.

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
