# Peer review of "Contrasting microbial assembly patterns in the woody endosphere of hybrid and non-hybrid Populus trees"

_PeerJ, doi:10.7717/peerj.20073_

## Round 0.1 · original submission · Major Revisions

Dear Dr. Grant,

Thank you for submitting your manuscript to PeerJ for publication. We are happy to inform you that your manuscript has been reviewed by two experts in your research areas and your manuscript has been recommended with major revisions before publication. The reviewers' comments are attached for your reference.

Please note that all raw microbial sequence data should be uploaded to a public database such as NCBI. Also, check all figures for accuracy.

Thank you for considering PeerJ for the publication of your research.

Best regards,

Tika Adhikari

Reviewer 1 ·

Basic reporting

The topic of this paper is the change of assembly patterns of endosphere microbial communities in two poplar species, but the research content is not consistent with the topic, and this paper only describes the diversity differences of endosphere microbial communities in two poplar species, including α diversity, β diversity and phylogenetic diversity. Differences in species composition are also described.

In conclusion, this paper shows very little research and it is recommended that the authors add a section on the function of endosphere microbial communities.

Experimental design

Methods were described with enough details.
Raw microbial data should be provided or uploaded to a public database such as NCBI.

Validity of the findings

The author does not refer to the community assembly model, which is divided into ecological and neutral processes, stochastic processes and deterministic processes according to different models. Authors should focus on community assembly models and give quantitative data based on these models.

Additional comments

L30 “suggesting more generalist bacterial associations”Please explain this sentence. Your introduction needs more detail. How did you reach this conclusion.

L34-35 Phylogenetic divergence led to more fungal lineages, but there was no significant difference in fungal diversity, and how did the authors arrive at this conclusion with more fungal associations?

Authors should be guided by scientific hypotheses, and presenting scientific hypotheses in the manuscript will make it easier for readers to understand the study

Fig.1 Please check whether the subgraphs are placed correctly. In Fig.1A, are both subgraphs ses.MPDab?

The attached figure and table should also be accompanied by a title and detailed annotations.

L278 “ however, interindividual taxonomic beta diversity was lower than phylogenetic (Fig. 2)” What is the significance of contrasting taxonomic beta diversity with phylogenetic together? How can this statement contribute to the research topic of this paper?

Reviewer 2 ·

Basic reporting

no comment

Experimental design

The research questions are not well defined, beyond the current statement about "examining" aspects of endophytic microbial communities. It is also unclear how the research fills an identified knowledge gap. The clarity of the manuscript would improve if the authors outline specific hypotheses or research questions in the Abstract and Introduction.

Validity of the findings

no comment

Additional comments

L17: The term “assembly” in this sentence is unclear. Consider framing it in the context of “community assembly”.

L21: The “x” in the hybrid name should be a times symbol not the letter “x” here and below.

L23: Ascomycete is not a formal term and shouldn’t be capitalized. Alternatively, use “Ascomycota”.

L27: Please clarify what the phrase “highly clustered phylogenetic structures” means.

L30-31: Please clarify what the phrase “terminal phylogenetic dispersion and interindividual beta diversity” means.

L34-36: This statement and conclusion is beyond the scope of this study which didn’t include any data specific to fungal pathogens. Can you provide a conclusion that is based on interpretation of results from this work?

L56-57: What do you mean by “abiotic and biotic factors”? These terms are very general. Can you provide specific examples from the citations?

L59-60: Please clarify what you mean by a “model” in this context. It may help to highlight why understanding ecological processes impacting endophytic communities is relevant beyond using this system as a “model”.

L61: Please clarify what “nearly ubiquitous environment” means in this sentence. There are many environments where phyllospheres are not present.

L70: This paragraph suggests phyllosphere microbiomes are primarily driven by deterministic processes. Are there examples from prior studies showing stochastic processes also play a role in community assembly?

L71: Please clarify what “eco-phylogenetic” approach means.

L75: Please provide a citation supporting the statement that Populus spp. are important successional species. Also, please reword this sentence to so the use of the genus “Populus” and “species” are in agreement.

L83-86: It would help to highlight your primary research questions or hypotheses here or somewhere else in the introduction.

L95: Should “hybrids” be referred to as “species”? I am not familiar with standards in plant taxonomy.

L202-203: I understand what is trying to be said in this sentence, but it reads a little oddly. Specifically, the phrase “phylogenetically related species are to each other” is unclear. All the species are phylogenetically related to each other.

L248: Please provide justification why taxa represented by fewer than 10 sequences were filtered from these data.

L252: The term “data” is plural.

L273: Please clarify what “interindividual taxonomic and phylogenetic beta diversity” means. What individuals are you referring to?

L322-323: Please clarify the phrase “ASVs that are distinct yet belong to specific lineages”.

L410: Do you mean “secondary metabolites”, rather than “secondary compounds”.

Figure 3. Please add a label to the x-axis. Also please include information in the legend on how data are aggregated.

---

## Round 0.2 · accepted · Accept

Dear Dr. Grant,

Thank you for submitting your manuscript to PeerJ for publication. I am happy to inform you that your manuscript has been recommended for publication and forwarded to the editorial office.

Congratulations, and thank you for considering PeerJ for the publication of your research.

Best regards,

Tika Adhikari

Reviewer 1 ·

Basic reporting

I have no other comments

Experimental design

the disign was done well. I have no other comments

Validity of the findings

I have no other comments

Additional comments

I have no other comments

Reviewer 2 ·

Basic reporting

no comment

Experimental design

no comment

Validity of the findings

no comment

Additional comments

The authors have addressed prior comments and improved the manuscript through inclusion of additional community assembly models and framing clear research questions.